# A New Implantation Method for Orthodontic Anchor Screws: Basic Research for Clinical Applications

**DOI:** 10.3390/biomedicines11030665

**Published:** 2023-02-22

**Authors:** Reiko Tokuyama-Toda, Hirochika Umeki, Shinji Ide, Fumitaka Kobayashi, Shunnosuke Tooyama, Mai Umehara, Susumu Tadokoro, Hiroshi Tomonari, Kazuhito Satomura

**Affiliations:** 1Department of Oral Medicine and Stomatology, School of Dental Medicine, Tsurumi University, 2-1-3, Tsurumi, Tsurumi-Ku, Yokohama 230-8501, Kanagawa, Japan; 2Department of Orthodontics, School of Dental Medicine, Tsurumi University, 2-1-3, Tsurumi, Tsurumi-Ku, Yokohama 230-8501, Kanagawa, Japan

**Keywords:** FEM analysis, implantation method, mini-screw, orthodontic anchor screws, von Mises stress

## Abstract

This study aimed to determine whether the positional relationship between the underside of the screw head and the surface of the alveolar bone could alter the stress on the two surfaces and affect the stability of implanted anchor screws. First, in order to confirm the extent of the gap between the mini-screw and the bone surface, a mini-screw was placed in the palate of rabbits and examined histologically. As a result, in the conventional screw implantation procedure, oral mucosa between the base of the screw head and the bone creates a spatial gap. Removal of the oral mucosa eliminates this gap. Then, we compared the positional difference of the screw in a contact and gap group by analyzing stress distribution on the bone and screw. Analysis using the finite element method showed that more stress was loaded on both the bone and screw in the gap group than in the contact group. Cortical bone thickness did not affect stress in either group. The effects of different load strengths were similar between groups. A surgical procedure in which mucosal coverings are removed so that implanted anchor mini-screws are in contact with the bone surface was found to reduce the stress load on both the bone and screw. This procedure can be used to prevent undesirable dislodgement of implanted mini-screws.

## 1. Introduction

Orthodontic anchor screws are an efficient and robust means of fixation in orthodontic treatment [1,2,3,4]. Since the mini-screw anchor system became widely used in clinical practice, it has been noted that there is occasional incidence of these screws falling out. To address this and develop means of prevention, several studies have investigated the biomechanics of mini-screw fixation. This has led to various improvements in the form of modifications to the shape [5,6,7] and length [8] of screws, consideration of bone condition at the fixation site [9,10,11,12,13], and optimization of screw angle and torque [14,15,16,17,18] during implantation. Nonetheless, mini-screws still fail in some cases, usually as a result of inflammation, infection, and shedding [19,20,21,22,23,24,25]. In this study, we investigated whether the position in which the screw is embedded affected its likelihood of falling out. Advantages of the mini-screw include its ability to penetrate the oral mucosa and its easy implantation into the bone, with no drilling required, making it a safe and convenient option [26,27,28]. However, when a mini-screw is implanted in the alveolar bone using this procedure, the oral mucosa lies between the underside of the screw head and the alveolar bone, causing a small separation between the screw and bone. When a load is applied in this situation, the alveolar bone is likely to be placed under more stress than it would if there were no gap between the screw and bone. Therefore, in this study, we used a rabbit mini-screw implantation model to compare the distance between the underside of the screw head and the alveolar bone surface, with and without intervening oral mucosa. The determined distances between the mini-screw and the bone surface were reproduced, and the distribution of stress on the alveolar bone and the mini-screw when clinically realistic orthodontic force was applied to the mini-screw was analyzed using the finite element method (FEM).

## 2. Materials and Methods

### 2.1. Animal Selection and Handling

Our animal care procedures and experimental protocols were approved by the Committee on the Ethics of Animal Experiments of Tsurumi University (permit number: 19A005). Surgical procedures were performed under sodium pentobarbital anesthesia, and all efforts were made to minimize the suffering of our laboratory animals. Ten male Japanese white rabbits (2.5–3.0 kg) were obtained from Tokyo Laboratory Animals Science Co., Ltd. (Tokyo, Japan). They were fed a normal diet and maintained under a 12 h light–dark cycle at 22 °C.

### 2.2. Implantation of Orthodontic Anchor Screws: Histological Examination to Confirm the Distance between the Bone Surface and the Mini-Screw

Mini-screws were implanted in the palatal bones of two groups of Japanese white rabbits to compare implantation preceded by removal of the oral mucosa with the conventional procedure without removal of the mucosa. In the removal group, the rabbits were given local anesthetic of 2% lidocaine. A 3 mm-diameter biopsy punch (Kai Industries, Gifu, Japan) was then used to remove a circular piece of the left palatal mucosa corresponding to the size of the screw head from each rabbit. A mini-screw was implanted in the same place without drilling, and it was confirmed that the initial fixation was good (*n* = 10). In members of the group that underwent the conventional procedure (the penetration group), 2% lidocaine local anesthetic was administered, and the mini-screw was then implanted in the right palate by penetrating the oral mucosa without drilling. It was confirmed that the initial fixation was good (*n* = 10). The mini-screw used with both groups had a diameter of 1.6 mm and a length of 5 mm (Jeil, Republic of Korea, Code 16-JK-005). The next day, maxillary tissue from around the mini-screw was obtained from each rabbit for histological analysis. The excised tissue was fixed with 10% neutral buffered formalin and embedded in methyl methacrylate (MMA) resin (Exakt, Germany). Polished 40–50 µm-thick sections were made and stained with hematoxylin and eosin (H&E).

### 2.3. Histometric Analysis

A histometric analysis was performed using the 10 tissue sections obtained from each group. The distances between the underside of the screw head and the bone surface were measured, and the average value for each group was calculated. The results were then analyzed using the FEM.

### 2.4. FEM Analysis Modeling

A three-dimensional bone block model with an integrated mini-screw was constructed using Optistruct v2020 (Altair Engineering, Troy, MI, USA), a computer-aided design support program. The cortical bone was simplified and simulated to a thickness of 1.0 mm or 2.0 mm, factoring in the buccal alveolar bone and palatal slope in which the mini-screw had been clinically placed. The shape of the mini-screw was set based on Jeil’s 16-JK-006 (Figure 1).

The interface between the cortex and the cancellous bone was assumed to be fully bonded, that is, to have continuous elements sharing the same nodes along the interface. A node-to-node contact condition was modeled on the interface between the mini-screw and the bone block to imitate a stage without osseointegration. All materials in the model were homogeneous, isotropic, and linearly elastic. The mini-screw was assumed to be pure titanium, with Young’s modulus of 110 gigapascals (GPa) and Poisson’s ratio of 0.33 [29]. For healthy bone quality, Young’s moduli of the cortical and cancellous bones were 18 GPa [29] and 1.37 GPa [8], respectively, and Poisson’s ratios were 0.3 for both (Table 1).

A static load was applied to the head of the mini-screw along the *x*-axis perpendicular to its long axis to simulate orthodontic force. The nodal solution of the von Mises stress in the bone and the mini-screw were calculated for each model with the FEM program. To determine the loading effects, two force magnitudes were applied to mimic various clinical conditions. A loading force of 2.0 Newtons (N) was used to mimic the load in space closure using a NiTi coil spring or elastomeric chains. A load of 10.0 N was used to mimic orthopedic force, such as mini-screw-assisted rapid palatal expansion. The load direction was set perpendicular to the long axis of the mini-screw. The distance between the underside of the screw head and the bone surface was set to 0 µm for the removal group and 765.6 µm for the penetration group based on the results obtained earlier (Table 2). FEM analysis was performed under the above conditions to examine the effects on the surrounding bone and on the mini-screw itself when the same force was applied to each group.

### 2.5. Statistical Analysis

All data were expressed as the mean ± standard deviation (SD). Statistical analyses were performed using the Kruskal–Wallis H test and Scheffe’s test. *p* values of <0.05 were considered statistically significant by using StatMate V.

## 3. Results

### 3.1. Distance between the Underside of the Screw Head and the Bone Surface Obtained from Histological Examination in Animal Experiments

The distances between the underside of the screw head and the bone surface in the removal and penetration group were compared. In the penetration group, there was an average gap of 765.6 µm. In the removal group, the underside of the screw head was in contact with the bone surface (0 µm) (Figure 2).

### 3.2. FEM Analysis

Figure 3 shows the peak von Mises stress on the bone at a load of 2.0 N. FEM analysis showed that the peak von Mises stress on the surrounding bone at 1.0 mm of the cortical bone was 3.3 MPa and 11.3 MPa in the removal and penetration group, respectively. Even with a cortical bone thickness of 2.0 mm, there was no significant difference in the peak von Mises stress value.

Figure 4 shows an aerial view of the screw head for this condition. It was found that the von Mises stress on the surrounding bone was lower in the removal group than in the penetration group.

Figure 5 shows the result of separating the von Mises stress into tension and compression. The compression was 3.9 MPa and 9.3 MPa in the removal and penetration group, respectively, indicating that, even with the same load of 2.0 N, the effect on the surrounding bone was much greater in the penetration group.

Figure 6 shows the peak von Mises stress on the mini-screw at a load of 2.0 N. The von Mises stress on the mini-screw itself was also lower in the removal group than in the penetration group.

The peak von Mises stress on the bone with a load of 10.0 N is shown in Figure 7. The peak von Mises stress on the surrounding bone at 1.0 mm of the cortical bone was 14.6 MPa and 54.8 MPa in the removal and penetration group, respectively. Even with a cortical bone thickness of 2.0 mm, there was no significant difference in the peak value of von Mises stress.

Figure 8 shows an aerial view of the screw head for this condition. As with the load of 2.0 N, it was found that the von Mises stress on the surrounding bone was lower in the removal group than in the penetration group.

Figure 9 shows the result of separating the von Mises stress into tension and compression. The compression was 17.7 MPa and 47.0 MPa in the removal and penetration group, respectively, indicating that, even with the same load of 10.0 N, the effect on the surrounding bone was greater in the penetration group.

Figure 10 shows the peak von Mises stress on the mini-screw at a load of 10.0 N. Again, the von Mises stress on the mini-screw was lower in the removal group than in the penetration group. These results are summarized in Figure 11.

## 4. Discussion

Since the mini-screw first came to be used as a fixation anchor in orthodontic treatment, various studies on the optimization of its implantation have been reported [5,6,7,8,9,10,11,12,13,14,15,16,17]. However, clinically undesirable cases in which screws drop out or break still occur, and the cause of this is unclear. In this study, we have focused on the gap between the underside of the screw head and the bone surface in the conventional implantation procedure and hypothesized that this gap may be a critical factor in the dropping out of screws. Using the FEM, we investigated the possibility that excessive stress results from the gap between the underside of the screw head and the bone surface, causing screws to fall out.

First, to confirm the existence of such a gap, mini-screws were implanted in the palatal bones of rabbits, with and without the removal of the oral mucosa at the site of screw placement before implantation. We found that, with the mucosa left in place, there was an average gap of 765 µm between the underside of the screw head and the bone surface. In the rabbits with the excised mucosa, no gap was present. By removing the local mucous membrane, the screw can be implanted so that the underside of the screw head comes into full contact with the bone surface.

The results of the FEM analysis are summarized in Figure 11. FEM analysis confirmed that there was greater von Mises stress in the penetration group than in the removal group with both a 2.0 N and 10.0 N load. The results were similar regardless of whether the cortical bone thickness was 1.0 mm or 2.0 mm. Concentration of stress was observed in the bone around the screw on the side to which the load was applied, and a stress peak was observed on the bone surface. When this stress was divided into tension and compression, greater compression was observed in the load direction and greater tension at 90° to the site. The tendency of these stresses was the same regardless of whether the load was 2.0 N or 10.0 N. However, the peak von Mises stress exceeded 50 MPa in the penetration group, especially at 10.0 N, suggesting that the extent of the stress is sufficient for bone resorption to occur. In comparison, the peak von Mises stress was suppressed to about 15 MPa in the removal group, in which the new implantation method proposed in this study was used. Even when the load was 2.0 N, the stress peak in the removal group was suppressed to about 1/4 of that seen in the penetration group. These results suggest that for screws with mucosal intervention, the greater the load, the greater the stress on the surrounding bone and the possibility of bone resorption. Thus, if the mucous membrane is removed and the screw is placed in contact with the bone surface, the stress on the surrounding bone is reduced and bone resorption is suppressed, preventing the screw from falling out.

The stress on the screw itself in the removal group was suppressed to about 1/2–1/3 of that seen in the penetration group at both 2.0 N and 10.0 N. In the penetration group, the stress to the screw itself was as large as 100 MPa at 10.0 N. Considering these results, in treatments where a large load is expected, such as mini-screw-assisted rapid expansion, it may be possible to help prevent screws from falling out by using screws with thicker diameters or embedding deeper.

The FEM analysis of the gap between the underside of the screw head and the bone surface was based on the results of an animal study, and the soft tissue thickness of human oral mucosa varies depending on the anatomical insertion site of the mini-screw [19,23,25,30,31,32], as well as the patient’s age and general condition [31,33]. Moreover, some changes in the surrounding mucosa and bone may occur due to the mechanical load. In light of these facts, the results in this study may not precisely reflect clinical conditions. Nonetheless, it is clear that soft tissue does intervene between the base of the anchor mini-screw head and the bone surface when it is implanted using the conventional procedure. It is also apparent that this can cause excessive stress to the surrounding bone tissue, leading to bone resorption and, eventually, to the screw falling out. Therefore, we strongly recommend the new procedure for implanting orthodontic anchor screws. The underside of the anchor screw should contact the bone surface directly, without oral mucosal intervention.

Indeed, the cause of the orthodontic anchor screw falling out is not limited to bone resorption. Screws sometimes fall out immediately after they are placed. The cause of these instances remains unclear. As previously reported, the cause may sometimes be infection or inflammation [19,20,21,22,23,24,25]. In this study, we proposed a new implantation method that can prevent mini-screws used as anchors in orthodontic treatment from falling out. Although further investigation is needed, this implantation method without mucosal intervention may be able to suppress excessive inflammation around the screw immediately after implantation. It is also possible that compression of the mucous membrane between the bottom of the screw head and the bone may cause the inflammation and infection of surrounding tissue. These issues should be investigated in future studies.

We compared the conventional mini-screw implantation method with the new implantation method, in which the mucosa was removed. We have shown that the new method can reduce various forces on bones. However, these studies were conducted using animal experiments and are only stress analyses by FEM. We believe that more research is needed before this new technique can be applied to clinical practice as a better alternative to traditional methods. Specifically, we believe that prospective studies should be conducted in two groups of patients: traditional and new. We are preparing to conduct a trial in the near future.

## 5. Conclusions

This study proposes a new procedure for implanting orthodontic anchor screws, in which the underside of the screw head is in direct contact with the bone surface, without mucosa between the two. This procedure helps to prevent excess stress on the surrounding bone and the loosening or falling out of screws.

## Figures and Tables

**Figure 1 biomedicines-11-00665-f001:**
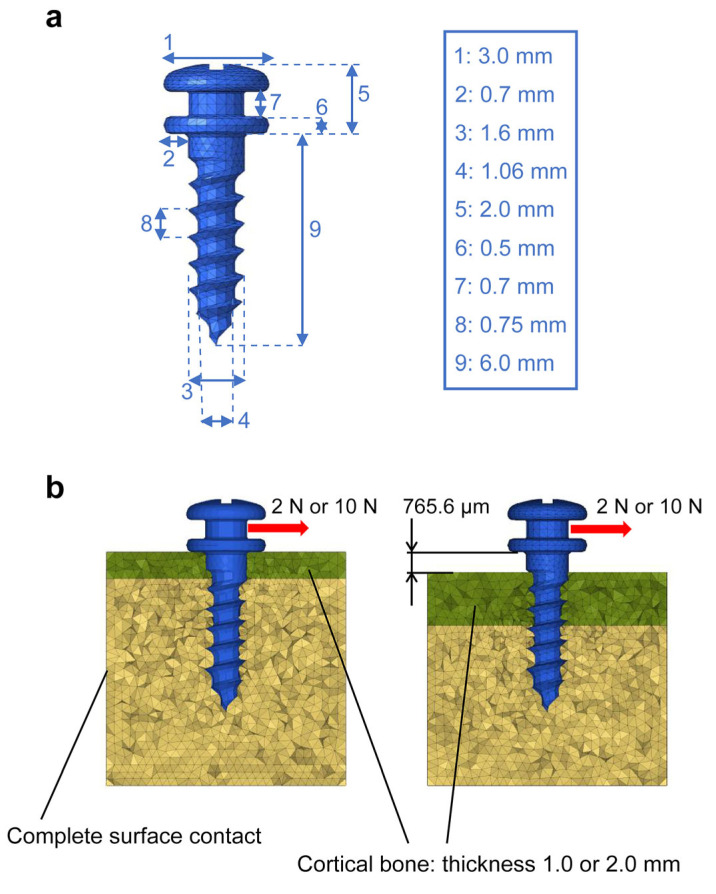
Setting the shape of an anchor screw shape and the conditions for finite element method analysis for a comparison of anchor screws with and without mucosa between the screw and bone. (**a**) Setting of the anchor screw shape. The shape of the mini-screw was set based on Jeil’s 16-JK-006. (**b**) Setting of the conditions for analysis using finite element method. Assuming the presence of the cortical and cancellous bone, the cortical bone was set to thicknesses of either 1.0 mm or 2.0 mm. The load was set to either 2.0 or 10.0 Newtons. The gap between the underside of the screw head and the bone surface was 0 µm in the removal group and 765.6 µm in the penetration group.

**Figure 2 biomedicines-11-00665-f002:**
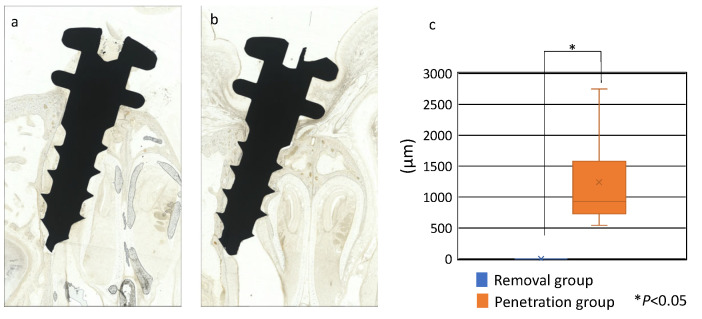
The distances between the underside of the screw head and the surface of the bone resulting from different methods of mini-screw implantation. (**a**) Removal group. (**b**) Penetration group. (**c**) The gap was an average of 0 µm in the removal group (*n* = 10) and 765.6 µm in the penetration group (*n* = 10).

**Figure 3 biomedicines-11-00665-f003:**
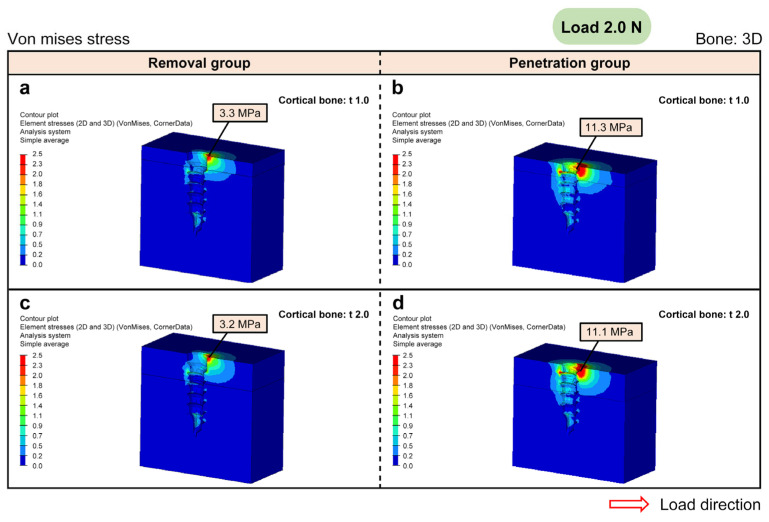
The peak von Mises stress on mini-screw-implanted bone with a load of 2.0 Newtons (3D view). (**a**,**c**) Removal group. (**b**,**d**) Penetration group. (**a**,**b**) Cortical bone thickness of 1.0 mm. (**c**,**d**) Cortical bone thickness of 2.0 mm. The peak von Mises stress on the surrounding bone with 1.0 mm of the cortical bone was 3.3 megapascals (MPa) in the removal group and 11.3 MPa in the penetration group. The peak von Mises stress on the surrounding bone with 2.0 mm of the cortical bone was 3.2 MPa in the removal group and 11.1 MPa in the penetration group.

**Figure 4 biomedicines-11-00665-f004:**
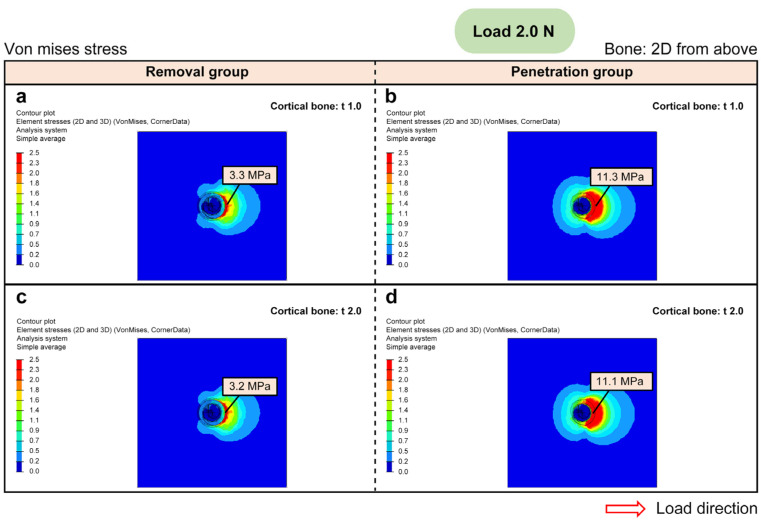
The peak von Mises stress on mini-screw-implanted bone with a load of 2.0 Newtons (2D view from above). (**a**,**c**) Removal group. (**b**,**d**) Penetration group. (**a**,**b**) Cortical bone thickness of 1.0 mm. (**c**,**d**) Cortical bone thickness of 2.0 mm. The peak von Mises stress on the surrounding bone with 1.0 mm of the cortical bone was 3.3 megapascals (MPa) in the removal group and 11.3 MPa in the penetration group. With 2.0 mm of the cortical bone, it was 3.2 MPa in the removal group and 11.1 MPa in the penetration group.

**Figure 5 biomedicines-11-00665-f005:**
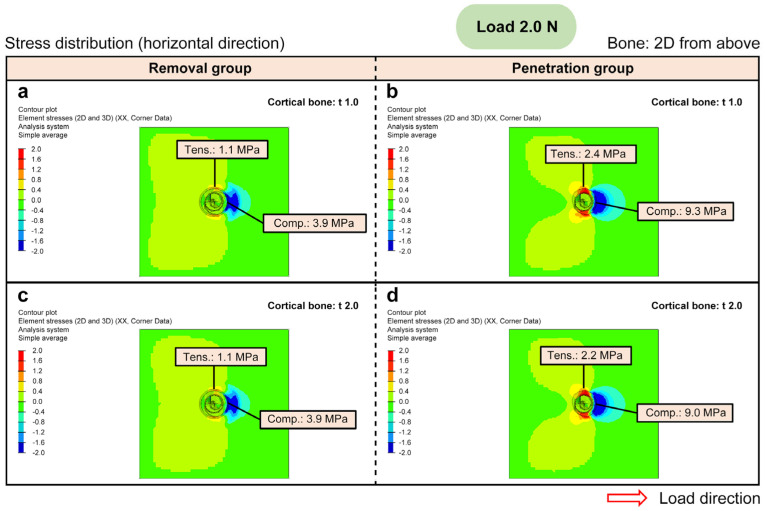
Deconstructing von Mises stress into the tension and compression on mini-screw implanted bone with a load of 2.0 Newtons (2D view from above). (**a**,**c**) Removal group. (**b**,**d**) Penetration group. (**a**,**b**) Cortical bone thickness of 1.0 mm. (**c**,**d**) Cortical bone thickness of 2.0 mm. With 1.0 mm of the cortical bone, the peak compression was 3.9 megapascals (MPa) and tension was 1.1 MPa in the removal group; compression was 9.3 MPa and tension was 2.4 MPa in the penetration group. With 2.0 mm of the cortical bone, the peak compression was 3.9 MPa and tension was 1.1 MPa in the removal group; compression was 9.0 MPa and tension was 2.2 MPa in the penetration group.

**Figure 6 biomedicines-11-00665-f006:**
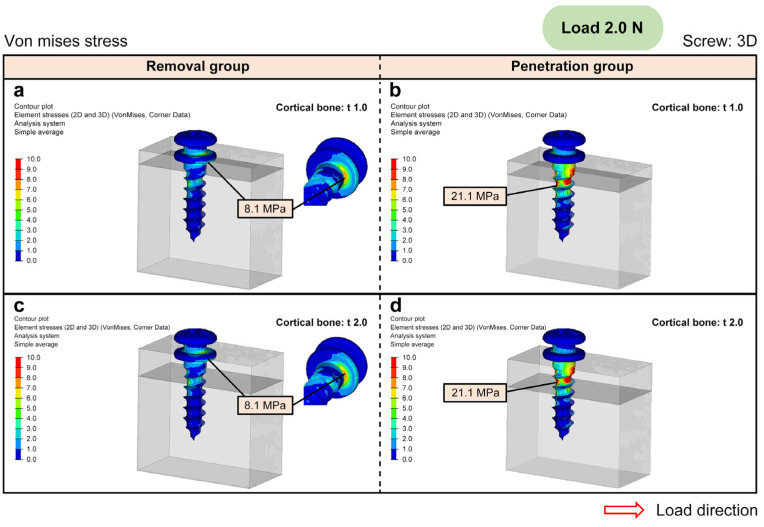
The peak von Mises stress for a mini-screw implanted in the bone with a load of 2.0 Newtons (3D view). (**a**,**c**) Removal group. (**b**,**d**) Penetration group. (**a**,**b**) Cortical bone thickness of 1.0 mm. (**c**,**d**) Cortical bone thickness of 2.0 mm. The peak von Mises stress on the mini-screw with 1.0 mm of the cortical bone was 8.1 megapascals (MPa) in the removal group and 21.1 MPa in the penetration group. The peak von Mises stress on the mini-screw with 2.0 mm of the cortical bone was 8.1 MPa in the removal group and 21.1 MPa in the penetration group.

**Figure 7 biomedicines-11-00665-f007:**
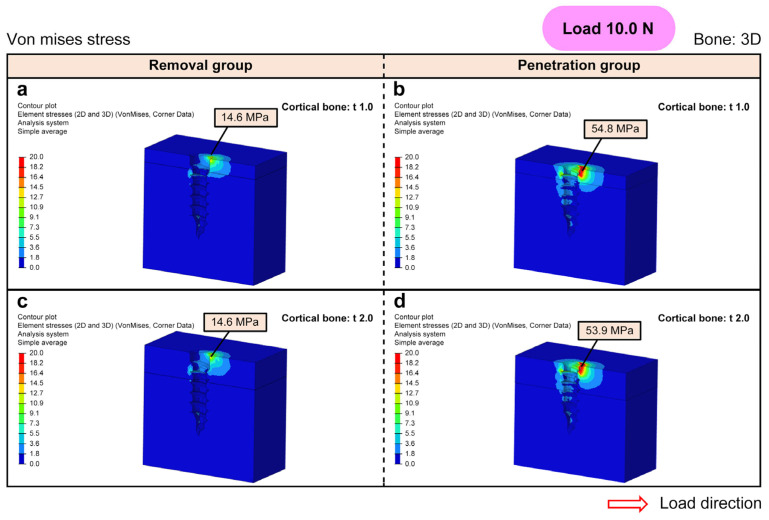
The peak von Mises stress on a mini-screw-implanted bone with a load of 10.0 Newtons (3D view). (**a**,**c**) Removal group. (**b**,**d**) Penetration group. (**a**,**b**) Cortical bone thickness of 1.0 mm. (**c**,**d**) Cortical bone thickness of 2.0 mm. The peak von Mises stress on the surrounding bone with 1.0 mm of the cortical bone was 14.6 megapascals (MPa) in the removal group and 54.8 MPa in the penetration group. The peak von Mises stress on the surrounding bone with 2.0 mm of the cortical bone was 14.6 MPa in the removal group and 53.9 MPa in the penetration group.

**Figure 8 biomedicines-11-00665-f008:**
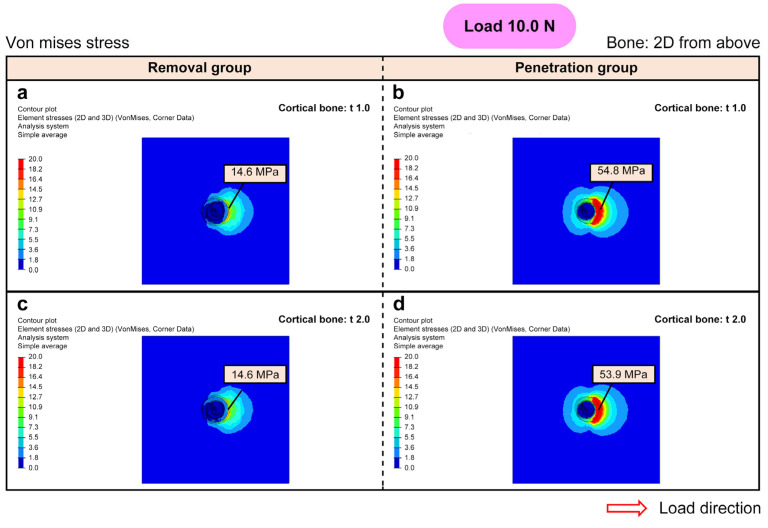
The peak von Mises stress on a mini-screw-implanted bone with a load of 10.0 Newtons (2D view from above). (**a**,**c**) Removal group. (**b**,**d**) Penetration group. (**a**,**b**) Cortical bone thickness of 1.0 mm. (**c**,**d**) Cortical bone thickness of 2.0 mm. The peak von Mises stress on the surrounding bone with 1.0 mm of the cortical bone was 14.6 megapascals (MPa) in the removal group and 54.8 MPa in the penetration group. The peak von Mises stress on the surrounding bone with 2.0 mm of the cortical bone was 14.6 MPa in the removal group and 53.9 MPa in the penetration group.

**Figure 9 biomedicines-11-00665-f009:**
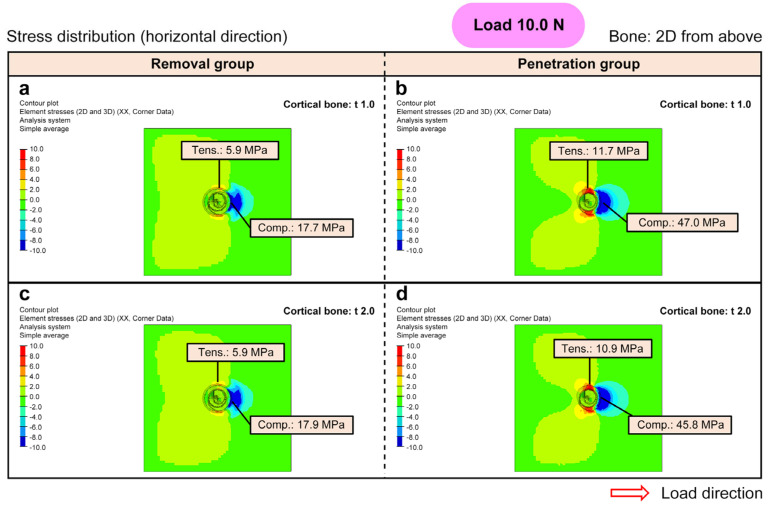
Deconstructing von Mises stress into the tension and compression on mini-screw implanted bone with a load of 10.0 Newtons (2D view from above). (**a**,**c**) Removal group. (**b**,**d**) Penetration group. (**a**,**b**) Cortical bone thickness is 1.0 mm. (**c**,**d**) Cortical bone thickness is 2.0 mm. With 1.0 mm of the cortical bone, the peak compression was 17.7 megapascals (MPa) and tension was 5.9 MPa in the removal group; compression was 47.0 MPa and tension was 11.7 MPa in the penetration group. With 2.0 mm of the cortical bone, the peak compression was 17.9 MPa and tension was 5.9 MPa in the removal group; compression was 45.8 MPa and tension was 10.9 MPa in the penetration group.

**Figure 10 biomedicines-11-00665-f010:**
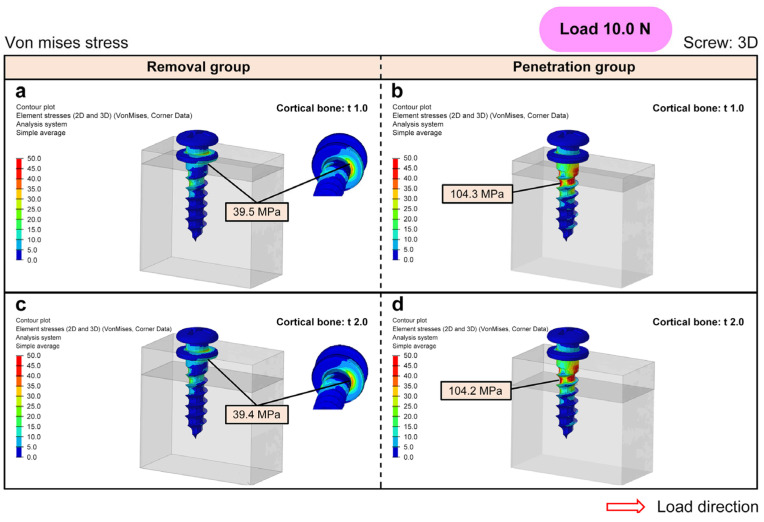
The peak von Mises stress on the mini-screw at a load of 10.0 N (3D view). (**a**,**c**) Removal group. (**b**,**d**) Penetration group. (**a**,**b**) Cortical bone thickness of 1.0 mm. (**c**,**d**) Cortical bone thickness of 2.0 mm. The peak von Mises stress on the mini-screw with 1.0 mm of the cortical bone was 39.5 megapascals (MPa) in the removal group and 104.3 MPa in the penetration group. The peak von Mises stress on the mini-screw with 2.0 mm of the cortical bone was 39.4 MPa in the removal group and 104.2 MPa in the penetration group.

**Figure 11 biomedicines-11-00665-f011:**
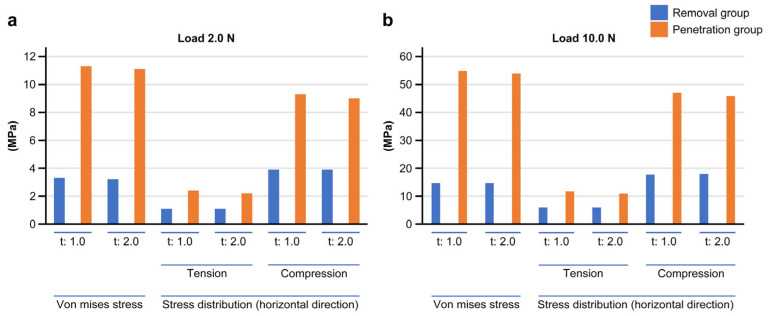
Summary of results of a comparison of mini-screw implantation into the bone with and without removal of the oral mucosa. (**a**) A load of 2.0 Newtons. (**b**) A load of 10.0 Newtons.

**Table 1 biomedicines-11-00665-t001:** Material property values.

Component	Material	Young’s Modulus [MPa]	Poisson’s Ratio	Density [ton/mm^3^]
Mini-screw	Titanium alloy	110,000	0.33	4.5 × 10^−9^
Cortical bone	—	18,000	0.3	0.9 × 10^−9^
Cancellous bone	—	13,700	0.3	0.9 × 10^−9^

**Table 2 biomedicines-11-00665-t002:** Analysis case list.

Case	Load [N]	Cortical Bone Thickness [mm]	Screw-Bone Distance [μm]		Element	Count
				Screw	Cortical Bone	Cancellous Bone
1–1	2.0	1.0	0.0	6601	17,957	124,262
1–2	2.0	1.0	765.6	6601	19,365	111,551
1–3	2.0	2.0	0.0	6601	38,807	88,182
1–4	2.0	2.0	765.6	6601	32,821	98,220
2–1	10.0	1.0	0.0	6601	17,957	124,262
2–2	10.0	1.0	765.6	6601	19,365	111,551
2–3	10.0	2.0	0.0	6601	38,807	88,182
2–4	10.0	2.0	765.6	6601	32,821	98,220

Note. Element type: Tetra-quadratic element.

## Data Availability

The data that support the findings of this study are available from the corresponding author upon reasonable request.

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
