# Peer review of "A New Implantation Method for Orthodontic Anchor Screws: Basic Research for Clinical Applications"

_biomedicines, 2023, doi:10.3390/biomedicines11030665_

Round 1
Reviewer 1 Report
Dear Authors,
We have read with interest your manuscript presenting a study about orthodontic mini screw. The concept is clear, with a well defined question (with clinical relevance).
The Methods are in adequation with the objective of the study, leading to precise results, and impacting conclusion.
We believe that the experiences conducted here might be of great interest for clinicians.
Therefore, we recommend to accept your manuscript as it is.
Yours faithfully,
Author Response
Feb 13, 2023
Dear Reviewer #1
Biomedicines
Re: Manuscript ID: Biomedicines-2194688
Type of manuscript: Article
Title: A New Implantation Method for Orthodontic Anchor Screws
Thank you for your valuable comments concerning our manuscript entitled " A New Implantation Method for Orthodontic Anchor Screws."
We have carefully studied your comments, made the necessary corrections, and are sending the revised manuscript here. The revised text has been colored.
Our responses to your comments are as follows:
Response to the comments of Reviewer #1
Dear Authors,
We have read with interest your manuscript presenting a study about orthodontic mini screw. The concept is clear, with a well defined question (with clinical relevance).
The Methods are in adequation with the objective of the study, leading to precise results, and impacting conclusion.
We believe that the experiences conducted here might be of great interest for clinicians.
Therefore, we recommend to accept your manuscript as it is.
Yours faithfully,
Response
Thank you very much for your encouragement concerning our manuscript.
I have added some sentences in the manuscript because other reviewers gave us some comments for the improvement of our manuscript.
We believe the manuscript has been improved satisfactorily and hope that it is now acceptable for publication in Biomedicines.
Yours sincerely,
Kazuhito Satomura, DDS, PhD
Reiko Tokuyama-Toda, DDS, PhD
Reviewer 2 Report
The paper "A New Implantation Method for Orthodontic Anchor Screws" is of some interest.
The abstract is clear and concise.
The introduction is clear enough; materials and methods are well described with many figures and tables to aid the reader's understanding. The results are clear and enriched by many (perhaps too much) tables. It needs to be summarized. The authors must underline the limitations of their study, which is absolutely to be considered a therapeutic proposal, conducted on a small number of rabbits. In the conclusions, authors should highlight the limitations of the study.
The study was carried out on only ten rabbits, and with a FEM analyses. The authors are very optimistic in their claims having conducted a study on a small sample and a FEM analysis.
Author Response
Feb 13, 2023
Dear Reviewer #2
Biomedicines
Re: Manuscript ID: Biomedicines-2194688
Type of manuscript: Article
Title: A New Implantation Method for Orthodontic Anchor Screws
Thank you for your valuable comments concerning our manuscript entitled " A New Implantation Method for Orthodontic Anchor Screws."
We have carefully studied your comments and made the necessary corrections, and we are sending here the revised manuscript. The revised text has been colored.
Our responses to your comments are as follows:
Response to the comments of Reviewer #2
The paper "A New Implantation Method for Orthodontic Anchor Screws" is of some interest.
The abstract is clear and concise.
The introduction is clear enough; materials and methods are well described with many figures and tables to aid the reader's understanding. The results are clear and enriched by many (perhaps too much) tables. It needs to be summarized. The authors must underline the limitations of their study, which is absolutely to be considered a therapeutic proposal, conducted on a small number of rabbits. In the conclusions, authors should highlight the limitations of the study.
The study was carried out on only ten rabbits, and with a FEM analyses. The authors are very optimistic in their claims having conducted a study on a small sample and a FEM analysis.
Response
Thank you very much for your comments.
The results are summarized in figure11. However, our description was inadequate and difficult to understand. I added the sentences for clarity (page 10, line 248).
Therefore, the number of rabbits would be sufficient for this experiment, which evaluated the difference in gap thickness between the two groups.
The study will also perform FEM analysis based on measurements of mini-screw and bone surface gaps obtained in animal studies. As you point out, we do not expect these results to lead to immediate clinical application of this technique. So we added the following statement:
"We compared the conventional mini-screw implantation method with the new implantation method in which the mucosa was removed. We have shown that the new method can reduce various forces on bones. However, these studies were conducted using animal experiments and are only stress analyses by FEM. We believe that more research is needed before this new technique can be applied to the clinical practice as a better alternative to traditional methods. Specifically, we believe that prospective studies should be conducted in two groups of patients: traditional and new. We are preparing to conduct a trial in the near future." (page 11, line 296, line -303)
We believe the manuscript has been improved satisfactorily and hope that it is now acceptable for publication in Biomedicines.
Yours sincerely,
Kazuhito Satomura, DDS, PhD
Reiko Tokuyama-Toda, DDS, PhD
Reviewer 3 Report
I would like to congratulate the authors for conducting the present study. I would like to share a few concerns:
The title should make reference to the type of study. In this case a finite elements method an histological analysis
Although the Abstract mentions briefly what was done, I believe they should be clearer on their methodological exposition in this part of the manuscript
I suggest the authors to place the keywords by alphabetic order
The results section 3.1 should somehow mention that these were the results of the histological analysis.
I recommend the authors to debate better the topics study limitations, strength, internal and external validity and results generalization
Author Response
Feb 13, 2023
Dear Reviewer #3
Biomedicines
Re: Manuscript ID: Biomedicines-2194688
Type of manuscript: Article
Title: A New Implantation Method for Orthodontic Anchor Screws
Thank you for your valuable comments concerning our manuscript entitled " A New Implantation Method for Orthodontic Anchor Screws."
We have carefully studied your comments and made the necessary corrections, and we are sending here the revised manuscript. The revised text has been colored.
Our responses to your comments are as follows:
Response to the comments of Reviewer #3
- The title should make reference to the type of study. In this case a finite elements method an histological analysis
Response
Thank you for your comment.
I have added the following to the title.
"A New Implantation Method for Orthodontic Anchor Screws: Basic Research for Clinical Applications"
- Although the Abstract mentions briefly what was done, I believe they should be clearer on their methodological exposition in this part of the manuscript
Response
Thank you very much for your comment.
I added a few sentences to the abstract. (page1, line13-15, 17)
- I suggest the authors to place the keywords by alphabetic order
Response
Thank you very much for your decision.
I sorted the keywords. (page1, line25, 26)
- The results section 3.1 should somehow mention that these were the results of the histological analysis.
Response
Thank you very much for your comment.
According to your comment, I added the following text:
“2.2. Implantation of Orthodontic Anchor Screws: histological examination to confirm the distance between the bone surface and the mini-screw” (page2, line60-61)
“3.1. Distance Between the Underside of the Screw Head and the Bone Surface obtained from histological examination in animal experiments” (page4, line128,129)
- I recommend the authors to debate better the topics study limitations, strength, internal and external validity and results generalization
Response
Thank you very much for your comment.
In compliance with your comment, I added some sentences to the manuscript:
"We compared the conventional mini-screw implantation method with the new implantation method in which the mucosa was removed. We have shown that the new method can reduce various forces on bones. However, these studies were conducted using animal experiments and are only stress analyses by FEM. We believe that more research is needed before this new technique can be applied to the clinical practice as a better alternative to traditional methods. Specifically, we believe that prospective studies should be conducted in two groups of patients: traditional and new. We are preparing to conduct a trial in the near future." (page 11, line 296-303)
We believe the manuscript has been improved satisfactorily and hope that it is now acceptable for publication in Biomedicines.
Yours sincerely,
Kazuhito Satomura, DDS, PhD
Reiko Tokuyama-Toda, DDS, PhD
Round 2
Reviewer 2 Report
The paper is worthy of publication now
Reviewer 3 Report
Dear authors, I have no more comments.